# Results of the National Contraception Survey Conducted by *Sociedad Española de Contracepción* (2020)

**DOI:** 10.3390/jcm11133777

**Published:** 2022-06-29

**Authors:** Fatima Leon-Larios, José Gutiérrez Ales, María José Puente Martínez, Marta Correa Rancel, Isabel Lahoz Pascual, Isabel Silva Reus, José Cruz Quílez Conde

**Affiliations:** 1Nursing Department, University of Seville, 41004 Sevilla, Spain; fatimaleon@us.es; 2Hospital Universitario Virgen Macarena, 41009 Sevilla, Spain; jgales050655@gmail.com; 3Hospital San Pedro, 26006 Logroño, Spain; 4Hospital Universitario de Canarias, 38320 Tenerife, Spain; tenerife1833@gmail.com; 5Departamento de Obstetricia-Ginecología, Pediatría, Preventiva, Medicina Legal y Forense, Microbiología, Parasitología, Universidad de la Laguna, 38200 Santa Cruz de Tenerife, Spain; 6Hospital Clínico Universitario Zaragoza, 50009 Zaragoza, Spain; isalahoz@yahoo.com; 7Centro de Salud Sexual y Reproductiva de Villena, 03400 Alicante, Spain; isasilreus@gmail.com; 8Hospital Universitario de Basurto, 48013 Bilbo, Spain; jotxe_c@hotmail.com

**Keywords:** contraception, sexual habits, women, long-acting reversible contraceptives, short-acting reversible contraceptives, counselling, emergency contraception

## Abstract

Background: The National Contraception Survey conducted by *Sociedad Española de Contracepción* intends to know the sexual and contraceptive habits of Spanish women of reproductive age. Methods: A descriptive and cross-sectional study with random sample selection was conducted with women aged from 14 to 49 years old in July and August 2020. Results: A total of 1801 women participated in the study, of which 78.7% used some contraception method during their sexual relationships. The most frequently used methods were condoms (31.3%) and combined oral contraceptives (18.5%) at their last sexual encounter. A total of 25.7% used both condoms and pills, especially younger women and those who had no steady partners (*p* < 0.001). Use of Long-acting Reversible Contraceptives continues to be low, although a slight increase in their use is observed, and they are recommended for 50% of the users who need contraception. Counselling on contraception was provided to 64.3% of the women, mainly by their gynaecologists. Regarding the decision to use a contraceptive method, the one suggested by health professionals was more influential, although this was not the case for women aged less than 20 years old (*p* < 0.001). A total of 38.4% of the women have used emergency oral contraception at least once and 66.8% of those who do not make continuous use of contraception methods do so out of personal choice. Conclusions: It is necessary to deepen work on counselling and awareness among the population towards the use of efficient contraceptive methods that prevent unplanned pregnancies.

## 1. Introduction

Age at sexual debut has gradually dropped in the new generations [1,2], which is one reason why it has become necessary to implement adequate sexual and contraceptive counselling and provide information that prevents unintended pregnancies and sexually transmitted infections [3]. The rate of unwanted pregnancies in Spain continues to be higher than desired [4]. According to the European Contraception Atlas, almost 4 out of 10 women undergo unplanned pregnancies [5], a reality that is similar to that of other countries such as France with 33.3% [2,6], Sweden with 22% [7] and the United Kingdom with one out of five pregnancies [3], although slightly lower than in the United States, where half of the women stated not having planned their pregnancies [8]. It is therefore deduced that a large part of the population does not have their contraceptive needs met. 

The unmet need for contraception presents high figures in Spain [5], exceeding those of other European countries such as France with less than 3% [9], Sweden with 8.9% [7] or Great Britain with 12.4% [3,10], although it is a more pressing problem in developing countries [11].

The use of the most frequent contraceptive methods has remained relatively unchanged throughout the years. Despite the recommendations, condoms and pills are still the most used methods worldwide [3,11,12,13]. It has been widely proved that Long-acting Reversible Contraceptives (LARCs) are effective in the prevention of unwanted pregnancies, abortion or recurrent abortions, although their rate of use is below desired [3,14,15]. Previous studies have already indicated the need to favour access to LARCs meeting women’s individual characteristics [16], regardless of factors such as education and income levels, marital status, or ethnicity [17,18]. 

Health education actions in terms of sexuality and contraception are increasingly implemented, and this can either exert an influence on the users’ behaviours or not [19,20,21]. A number of already published studies indicate that the decision to use a given contraceptive method over another is usually influenced by the advice of a health professional [22,23]. In addition, there are factors that can exert an influence on the choice and use of one method over another. Some of these factors are education level, income level and religious beliefs, among others [20,24,25]. Therefore, it is necessary to consider them when individualized counselling is provided to the user. 

Counselling is offered during consultations by nurses, midwives, primary care physicians and gynaecologists, although only the latter two are authorized to write medical prescriptions [22]. Access to contraceptive methods in Spain is affordable, considering that some of them are fully funded by the National Health System. However, access to counselling on contraception and sexual education are not uniformly egalitarian throughout the country, as they depend on regional policies [19]. Thus, while they are practically fully funded in some regions, there are deficits in others [26].

The main objective of this study was to understand the habits of the Spanish female population in relation to the use of contraceptive methods and sexual health. We are interested in determining the factors that favour the use of some methods over others in order to design strategies that allow an increase in the use of contraceptive methods and thus avoid unplanned pregnancies. 

## 2. Materials and Methods

### 2.1. Study Design

A descriptive and cross-sectional study. This study is an analysis of the National Contraception Survey conducted by *Sociedad Española de Contracepción* every 2 years through SigmaDos, an international Marketing and Public Survey Study company headquartered in Spain. 

### 2.2. Measures

An ad hoc questionnaire designed by health professionals specialized in contraception who are members of *Sociedad Española de Contracepción* was used. It includes data on the use of contraceptive methods as well as about the participants’ sexual habits. 

### 2.3. Data Collection

The survey was conducted via telephone calls by means of a structured questionnaire consisting of 31 questions related to the interviewees’ sexual and reproductive health. It was conducted from 31 July to 15 August 2020.

### 2.4. Sample Size

The women included were those of reproductive age between 14 and 49 years old throughout Spain. Sample distribution was proportional to that of the Spanish actual population, with an adjustment system by age quotas, in which the last interviewee selection instance was fully random. This distribution allowed offering representative data at the national level, although not at the level of geographical regions or areas.

Sample size was 1801 participants, with a possible error of +2.35% for a 95.5% confidence level. 

### 2.5. Data Analysis

A univariate and bivariate analysis was performed to evaluate and know sexual behaviours and use of contraceptive methods in women of reproductive age. The χ^2^ test was used to understand the relationship between use of the contraceptive methods and the sociodemographic characteristics, reporting φ as an indicator of size effect. The chi-square test was used to understand the relationship between use of the contraceptive methods and sociodemographic characteristics. A multivariate logistic regression was performed to find out the relationship between different variables. The results of the regression analysis were presented in terms of adjusted Odds Ratio (OR_adj_), with a 95% confidence interval and significance level. Interpretation of the adjusted Odds Ratio can be as follows: if OR_adj_ > 1, then there are higher odds of using contraceptives.

In all hypothesis contrasts, the significance level was set at 0.05. Data analysis was processed in IBM SPSS 26 (IBM Corp., Armonk, NY, USA). 

### 2.6. Ethical Aspects 

This questionnaire was answered anonymously and ensuring the confidentiality of the participants. The women were free to stop answering questions at any moment. The Spanish health authorities do not require approval by a Research Ethics Committee for this type of study in which the participants are asked to provide data about their sexual and contraceptive practices outside a health environment that establishes a professional relationship. Verbal informed consent was requested from the participants at the beginning of the study. 

## 3. Results

A total of 1801 women took part in the study, with a final response rate of 98.72%, which represents 1778 women. The participants’ sociodemographic characteristics are presented in Table 1.

### 3.1. Sexual Habits 

In total, 95.46% of the women aged between 15 and 49 years old had already engaged in sexual relationships at least once (16 years old is the minimum age for sexual consent). Globally, the mean age at the first sexual relationship was 18.07 years old. The age at the time of their first sexual relationship fluctuates between 19.83 years old in the case of the women aged between 45 and 49 years old and 15.83 years old in those aged between 15 and 19, as can be seen in Figure 1.

Of the total of participating women, 89.8% engaged in sexual relationships with one of the frequencies shown in Figure 2, with determination that the higher the age, the lower the monthly frequency of sexual relationships: χ^2^(18) = 61.56, *p* < 0.001, φ = 0.190.

Out of all the women, 91.9% stated that they had engaged in sexual relationships in the last 12 months; on the other hand, 3.7% had already had sexual relationships at some moment in their life, although not in the last year. The mean number of sex partners during the last year was 1.36, being slightly higher among women aged less than 30 years old, as follows: from 15 to 19 years old, 1.71 partners; from 20 to 24 years old, 1.53; and from 25 to 29 years old, 1.51.

#### Sexual Relationships and Use of Contraceptive Methods

When the women were asked if they have engaged in sexual relationships without contraceptive protection, 61.4% answered never and 15.8%, almost never. 21.3% reported having had sexual relationships with no contraceptive protection quite frequently, with a higher percentage in the age group from 30 to 34 years old, reaching 26.8%. Likewise, those women that lived with their partners (26.8%) and were Spanish (21.7%) were more likely to engage in sexual relationships without using any contraceptive method: χ^2^(4) = 12.64, *p* = 0.013, φ = 0.091.

The women at risk of unwanted pregnancies represented 22.9% of those that did not use any contraceptive method, accounting for 6.2% of the total women of reproductive age. 

### 3.2. Use of Emergency Oral Contraception 

In total, 38.4% of the women of reproductive age interviewed stated having consulted at some point about emergency oral contraception products that are dispensed free of charge in pharmacies at some moment. This percentage is higher in the age group from 20 to 39 years old, as shown in Figure 3. 

Use of the morning-after pill is more frequent in women that do not live with their partners (46.1%) [χ^2^(4) = 23.51, *p* < 0.001, φ = 0.118] or do not have a steady partner (43.4%) [χ^2^(4) = 78.88, *p* = 0.008, φ = 0.216], as well as among those who resort to condoms as their usual contraceptive method (46.8%) [χ^2^(15) = 25.59, *p* = 0.043, φ = 0.153], as shown in Table 2.

### 3.3. Information and Counselling on Contraceptive Methods 

Most of the women (64.3%) consulted their primary care physician, gynaecologist, midwife or Family Planning Centre to receive counselling and therefore choose the method that best fitted their personal circumstances. The women belonging to the oldest age group attended health centres more frequently to receive individualized counselling: χ^2^(12) = 83.39, *p* < 0.001, φ = 0.217). It was noticed that the youngest women attended the centres in a lower proportion (37.8% of those aged less than 20 years old). 

The contraceptive methods used by the women were as follows: the one indicated by the gynaecologist (42.4%), self-prescription (39.5%), the one suggested by the family physician (12.1%), a midwife/nurse (1.7%) and a pharmacist (0.6%). When the women were asked who was most influential on the definite choice of the method used, 44.2% stated that it was the health personnel, 31.2% indicated that it was their own decision, in 17.2% it was family members/friends, 2.4% mentioned the communication media, 1.9% pointed to social networks/Internet, 1.9% chose Others and 1.3% selected Does Not Know/Does Not Answer. 

The health staff exerts less influence on the final decision about using one method over another, especially in those aged less than 20 years old, among which family members and friends are the most influential people (40.4%). If we perform an analysis according to the method used, it is observed that the women who chose oral combined hormonal contraception (69.3%) or IUD (60.9%) were driven by the influence exerted by a health professional. Personal choice for their use (42.3%) predominated among those who used condoms.

Not attending the consultation to receive counselling was not related to the schooling and income levels or to the religious beliefs but to nationality: χ^2^(2) = 8.22, *p* = 0.016, φ = 0.068, as can be seen in Table 3.

#### Information and Counselling about Long-Acting Reversible Methods

In total, 49.9% of the women aged between 15 to 49 years old stated having been advised and offered information about some long-acting method (IUD or implant) at some moment. It was observed that it is more frequently offered to the oldest age group: χ^2^(12) = 44.65, *p* < 0.001, φ = 0.158, as can be seen in Figure 4.

When the women were asked about the advantages perceived in using long-acting contraception methods, convenience was signalled as the main advantage of their use. 47.3% of the women aged between 15 and 49 years old identified this quality. Contraceptive efficacy (26.8%) was identified in the second place, while the greater safety offered by these contraceptive methods was ranked third (25.1%).

### 3.4. Contraceptive Use Pattern

It was found that 7 out of 10 Spanish women of reproductive age used some contraceptive method. It is to be considered that, among the women who did not resort to any contraceptive method are those that did not engage in sexual relationships while the study was conducted.

The age group that made the most use of some method during sexual relationships is the one from 20 to 30 years old, as can be seen in Figure 5. 

In relation to other analysis variables, it is worth noting the increase in the number of women who did not use any contraceptive method among those with lower income levels (33.9%) and those who are Catholics (32.2%); with this rate increasing especially with women who already have children and want more (41.8%): χ^2^(6) = 28.50, *p* < 0.001, φ = 0.130.

Condoms are the contraceptive method most frequently used (31.3%) by the women of reproductive age who engage in sexual relationships and use some method. Combined oral hormonal contraception ranks second (18.5%), followed by copper IUD (4.3%) and hormonal IUD (4.2%). Table 4 presents the contraceptive methods used by age groups from the total of women between 15 and 49 years old participating in the study. Condoms continue to be the most frequently used method in all age groups, and pills reach their maximum use value between the ages of 20 and 24, subsequently decreasing as age advances. 

Considering the different contraceptive methods and their characteristics, it is worth noting that 69.6% of the women of reproductive age used effective methods, reaching 99% of those that consulted about some method. 31.3% used barrier methods, 23.8% consulted regarding hormonal methods and 8.3% employed some type of IUD, as shown in Table 4. 

#### 3.4.1. Drivers for the Use of Reversible Methods

Overall, the reasons stated by the women for using the short-acting reversible contraceptive methods (SARCs) were as follows: convenience (48.4%) for all age groups, safety (37.5%), lower impact on health (37.5%), lifestyle (16.1%), medical recommendation (14.4%), contraceptive efficacy (13.1%), economic reasons (3.1%), DNK/DNA (1.9%) and Others (1%).

#### 3.4.2. Hormonal Methods

The hormonal methods were employed following medical recommendations (23.9%) and for regulation of the menstrual cycle (19.8%). Those women who showed reluctance to use hormonal methods argued risk of hormonal use (29.1%), side effects and/or health problems (28.7%).

#### 3.4.3. Discontinuation of Hormonal Contraception

In total, 68.9% of the participating women that sought consultations about combined hormonal methods did not discontinue their use periodically. On the other hand, less than one-third did so, due to personal choice (66.8%) based on diverse information or recommendations. In total, 29.7% implemented periodic discontinuation due to the health recommendations provided by physicians, midwives or nurses. Discontinuation tends to be higher with advancing age, exceeding 30% from the age of 25. The women who used pills interrupted their contraception more often over lifetime [χ^2^(12) = 139.11, *p* < 0.001, φ = 0.358], as well as those that already had children [χ^2^(4) = 50.98, *p* < 0.001, φ = 0.381], as can be seen in Table 5.

#### 3.4.4. Use of the Dual Method

Of all the women of reproductive age that use contraceptive methods, 25.7% stated resorting to the dual method (condoms and hormonal method and/or IUD). The most usual dual method combination corresponded to that of condoms and pills (28.1%). This practice was more frequent among the youngest women (from 15 to 19 years old). 8.8% consulted about it occasionally (almost never/sometimes) and 16.9% did so always/almost always.

Habitual use of the dual method (*always/almost always*) was more frequent among the women who had no steady partners (26.7%): χ^2^(10) = 60.51, *p* < 0.001, φ = 0.235, as indicated in Table 6.

## 4. Discussion

This study intended to know the sexual behaviours and use patterns of contraceptive methods of women of reproductive age living in Spain. To such an end, this descriptive and cross-sectional study was conducted with a random and stratified sample of Spanish women of reproductive age in 2020. 

Spanish women’s age at the time of their first sexual relationship is estimated to be less than 18 years old. However, it is noticed that such age has been dropping according to the age groups. Women aged less than 20 years old initiated their sex lives four years earlier than their mothers, a finding that is in line with other European studies that estimate the age at the time of first sexual relationship to be 16 years old [2,7]. We have to be aware of the fact that women initiate their sex lives increasingly earlier in time; therefore, they have to resort to proper contraception methods if they want to avoid sexually transmitted infections or unwanted pregnancies; awareness of these methods can be achieved through sexual education [19,21]. 

The results obtained in this National Survey indicate that 7 out of 10 women use some type of contraceptive method. This is similar to the data contributed by other European studies [3,7], in which it was also already indicated that the higher the user’s age, the lesser the use of contraceptive methods [27]. If we consider that the women that are at risk of unwanted pregnancies are those who, with the possibility of having children, do not want them and engage in sexual relationships without resorting to any contraceptive method, we would be talking about 2 out of 10 women. This tells us that, for this reason, it is still necessary to deepen work on prevention, especially in the extreme age groups: below 20 years old [28] and above 45 years old, which are the age groups where the highest use inconsistency is observed. More influence should be exerted on collective groups that respond to these premises, as counselling interventions turn out to be effective to increase contraceptive use and compliance rates [19,29].

Among the contraceptive methods most frequently used in women of reproductive age, condoms still rank first, followed by pills, although not only in Spain [12], as we notice the same trend if we investigate data published in other European countries [7,13]. Despite the increase in the use of hormonal methods, condoms are still the most frequent choice. However, their use decreases when women start to consider their partners as stable, a fact that has already been identified in similar studies [30]. The dual method is used, but especially in younger women, as the use of condoms is reduced as age advances [21]. Inconsistent condom use is related to late placement after intercourse has started; better condom use is noticeable in the youngest women (aged less than 19 years old), a fact that has not always been found in similar studies [31].

Counselling in terms of contraception is usually in charge of health professionals (family physicians, gynaecologists, midwives, nurses), although it is still more common for older women to attend contraceptive counselling appointments [32], reason why reinforcement is required in the youngest women regarding the need to receive information and guidance by duly qualified professionals, as already pointed out in previous studies [19,22]. The woman’s choice regarding one method or another is influenced by the prescribing health professional, followed by her own personal criterion [22,33,34].

There is still a considerable number of women who resort to emergency oral contraception at some moment during their reproductive life, especially among those aged less than 40 years old [35]. Almost 4 out of 10 women have made use of emergency oral contraception at least once, a result that is in consonance with others published with young people belonging to similar age groups [4,36,37]. As was already pointed out by some studies, using it after a failure in the usual contraceptive method might contribute to reducing the number of unplanned pregnancies [4].

Despite its recommendation and of being a more economical option [26], the use of LARCs continues to be relatively low when compared to SARCs, a fact that has also been stated in studies conducted in other European countries such as Sweden, where 2 out of 10 women used LARCs [7]. It continues to be a more frequent contraception option in women aged over 30 years old [3].

Nevertheless, it was observed that counselling and the information provided about LARCs had increased, leading to an increase in their use when compared to previous years [3,13,38]. In addition, the recommendation to use them among the youngest women is increasingly frequent [39], as pointed out in our study. It is important that the family physicians and gynaecologists that will prescribe them know them, and be aware of and minimize personal bias for their recommendation [40]. In addition, when health professionals have sound knowledge of the method, they will better convey the information to the users, who will be able to choose it on better grounds. It has been shown that, when there are knowledge gaps, its use is lower among women [15,22], whereas when they know it in detail and know of someone who uses it, they consider it highly acceptable [41].

Among the main drivers for its use are safety and convenience, as well as prevention of unwanted pregnancies, as already pointed out by women participating in other similar studies [7]. Access and policies are usually important for the use of LARC methods. Those countries that have implemented policies that ensure availability and insertion of the device free of charge increase their use among women [16,42]. In our study, it has been noticed that many users underwent annual IUD control reviews, despite not being a specific recommendation [43], being more frequent in women with lower schooling levels. 

On the other hand, a clear reduction is noticed in the habit of periodically discontinuing the use of hormonal methods. In 2014, 51% of the users who consulted about pills implemented intervals, with a progressive reduction until 2020, currently reaching 28%. Likewise, a difference is noticed in relation to previous years; it now responds to the women’s own decisions rather than to a recommendation by a health professional [44]. This is in line with other studies conducted where a reduction in discontinuation of hormonal contraception was observed, although it is still used despite not being recommended, reason why work should be deepened to properly advise women [45].

## 5. Strengths and Limitations

One of the strengths of this study is the sample size achieved, which provides additional information with women from all regions and age groups, therefore allowing us to observe a real trend at the national level. As this is a cross-sectional study, we cannot establish causality, only associations between the variables defined. However, this study allows us to have a solid baseline of the reality regarding women’s use of contraception and sexual behaviours in Spain. 

This study was conducted during the pandemic that we are currently experiencing, therefore, its results must be interpreted considering this scenario. The results contributed by this National Contraception Survey were obtained during 2020; consequently, they are influenced by the situation experienced during the last two years in Europe after the declaration of the global pandemic, which may have affected the results.

## 6. Conclusions

The results found in this research allow focusing the attention on those women who are susceptible to not using consistent contraceptive methods. It allows defining the strategies for counselling by health professionals for the proper use of contraceptive methods that allow preventing unwanted pregnancies and abortions. It is necessary to continue developing strategies that reduce the unmet need for contraception in order to avoid unwanted pregnancies and continue working to identify the gaps that will allow for efficient counselling and adherence in the users, meeting their individual needs. It becomes necessary to keep fostering the use of LARCs in all age groups, not only in those women with their desire to conceive fulfilled or in those who belong to older age groups.

## Figures and Tables

**Figure 1 jcm-11-03777-f001:**
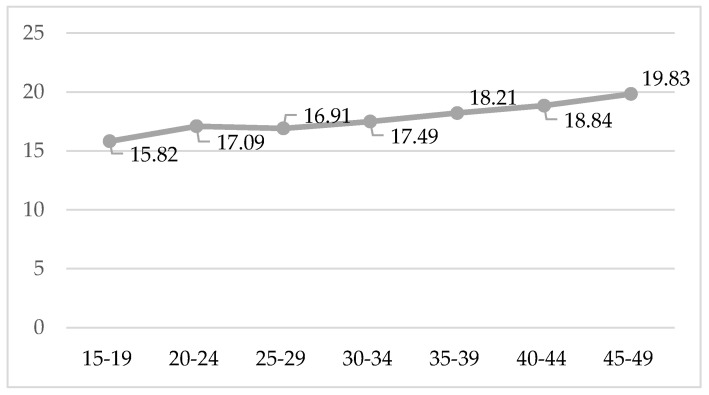
Age at the first sexual relationship according to age group.

**Figure 2 jcm-11-03777-f002:**
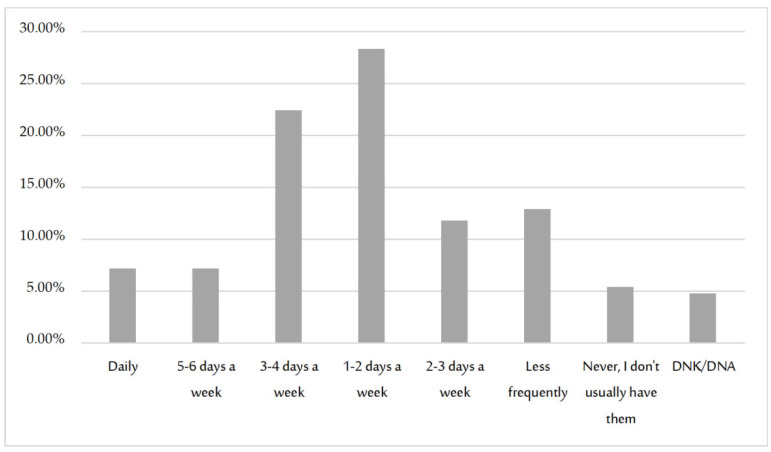
Frequency of sexual relationships.

**Figure 3 jcm-11-03777-f003:**
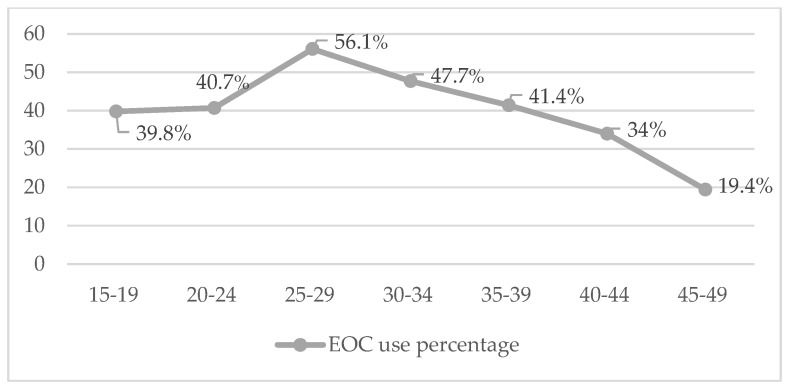
Use of the morning-after pill according to age group.

**Figure 4 jcm-11-03777-f004:**
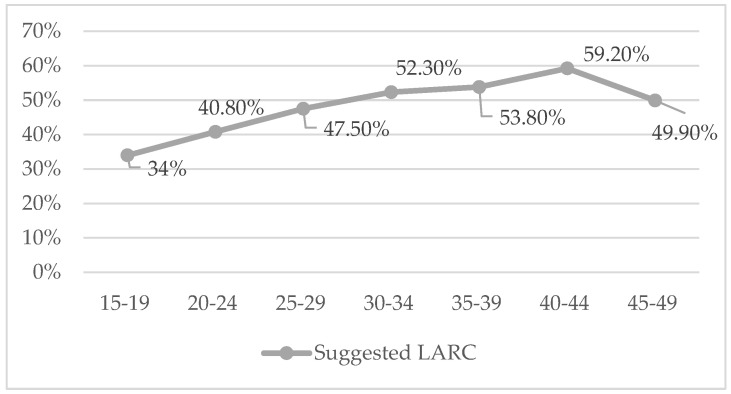
Counselling for the use of LARCs according to age group.

**Figure 5 jcm-11-03777-f005:**
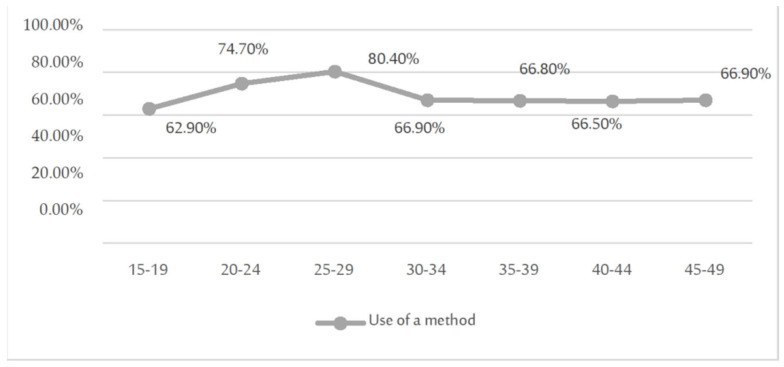
Use of some contraceptive method according to age group.

**Table 1 jcm-11-03777-t001:** Sociodemographic characteristics.

Variables	Total *n* (%) 1778 (100)
Age	
15–19 years old	170 (9.6)
20–24 years old	189 (10.6)
25–29 years old	211 (11.9)
30–34 years old	249 (14)
35–39 years old	308 (17.3)
40–44 years old	335 (18.8)
45–49 years old	316 (17.8)
Relationship situation	
Lives with her partner	1011 (56.9)
Steady partner, but not living together	325 (18.3)
No steady partner	442 (24.9)
Schooling level	
Elementary School	84 (4.7)
High School	917 (51.9)
University Studies	767 (43.4)
Nationality	
Spanish	1602 (90.2)
Other	174 (9.8)
Income level	
Low	401 (25.8)
Average-Low	663 (42.6)
Average	300 (19.3)
High	191 (12.3)
Work situation	1748 (100)
Working	1036 (59.3)
Not working	711 (40.7)
Religious beliefs	
Practicing Catholic	242 (13.6)
Lapsed Catholic	753 (42.4)
Other religions	111 (6.2)
Agnostic/Atheist	673 (37.8)

**Table 2 jcm-11-03777-t002:** Use of Emergency Contraception Pills (ECPs) over lifetime according to usual contraceptive method, nationality and relationship status.

Use of ECPs	Total	Relationship Situation			Contraceptive Methods Generally Used						Nationality	
		Lives with Her Partner	Steady Partner, Not Living Together	No Steady Partner	Condoms	Pills	IUD	Voluntary Sterilization	Other Methods	None	Spanish	Other
Yes	38.4%	34.1%	46.1%	43.4%	46.8%	40.8%	37.7%	30.0%	46.4%	30.2%	38.3%	39.8%
No	61.5%	65.9%	5.9%	56.3%	53.2%	59.2%	62.3%	70.0%	53.6%	69.8%	61.7%	60.2%
DNK/DNA	0.1%	0.0%	0.0%	0.3%	0.0%	0.0%	0.0%	0.0%	0.0%	0.0%	0.0%	0.0%

**Table 3 jcm-11-03777-t003:** Counselling on contraception over lifetime according to nationality, schooling and income levels and religious beliefs.

Counselling	Total	Nationality		Schooling Level			Income Level				Religious Beliefs			
		Spanish	Other	Elementary School	High School	University Studies	Low	Average-Low	Average	High	Practicing Catholic	Lapsed Catholic	Professes Other Religions	Agnostic/Atheist
Yes	64.3%	65.5%	54.6%	57.7%	64.2%	65.8%	63.3%	66.7%	71.4%	62.7%	62.6%	65.7%	59.3%	64.3%
No	35.6%	34.5%	45.4%	42.3%	35.8%	34.2%	36.7%	33.3%	28.6%	37.3%	37.4%	34.2%	40.7%	35.7%
DNK/DNA	0.0%	0.0%	0.0%	0.0%	0.1%	0.0%	0.0%	0.1%	0.0%	0.0%	0.0%	0.1%	0.0%	0.0%

**Table 4 jcm-11-03777-t004:** Distribution of the use of contraceptive methods.

Effective methods (98.5%)	Barrier	Condoms		47.5%
	Hormonal (36%)	Oral administration (29.1%)	Pills	28.1%
			Mini pills	1.0%
		Intravaginal administration	Vaginal ring	3.2%
			Patch	1.0%
		Injectable (monthly-quarterly)		0.5%
		Subcutaneous implant (1 or 2 bars)		2.2%
	IUD (12.5%)	Copper		6.5%
		Hormone release		6.0%
	Voluntary sterilization (irreversible) 2.5%	Fallopian tube ligation/Essure procedure (tubal obstruction)		1.6%
		Vasectomy		0.9%
Little effective methods (0.2%)	Natural or Periodic abstinence	Knauss–Ogino (standard day method), Billings’ method (cervical mucus), temperature method		0.1%
	Interruption	Coitus interruptus (withdrawal)		0.1%
Others				0.7%

**Table 5 jcm-11-03777-t005:** Discontinuation in the use of contraceptives according to nationality, schooling level and desire to conceive fulfilled.

Discontinuation	Total	Nationality		Schooling Level			Motherhood Situation			
		Spanish	Other	Elementary School	High School	University Studies	Has Children and Wants More	Has Children and Does Not Want More	Does Not Have Children but Wants to	Does Not Have Children and Does Not Want to
Yes	28.0%	28.7%	15.7%	19.2%	29.5%	26.5%	31.8%	31.5%	26.6%	26.2%
No	68.9%	68.2%	80.1%	75.9%	68.4%	69.0%	68.2%	65.7%	69.7%	70.3%
DNK/DNA	3.2%	3.1%	4.2%	4.9%	2.2%	4.5%	0.0%	2.8%	3.7%	3.5%

**Table 6 jcm-11-03777-t006:** Use of the dual method according to nationality, having a partner or not and schooling level.

	Total	Nationality		Schooling Level			Relationship Situation		
		Spanish	Other	Elementary School	High School	University Studies	Lives with Her Partner	Steady Partner, but Not Living together	No Steady Partner
Yes (Always/Almost always)	16.9%	17.1%	14.6%	22.6%	18.8%	14.6%	11.8%	20.5%	26.7%
No (Never, Almost never or Sometimes)	82.6%	82.4%	85.4%	75.1%	81.0%	84.8%	87.7%	79.5%	72.6%
DNK/DNA	0.4%	0.5%	0.0%	2.3%	0.1%	0.6%	0.5%	0.0%	0.7%

## Data Availability

The datasets that support the findings of this study are available from the corresponding author upon reasonable written request.

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
