# Peer review of "Results of the National Contraception Survey Conducted by Sociedad Española de Contracepción (2020)"

_jcm, 2022, doi:10.3390/jcm11133777_

Round 1
Reviewer 1 Report
abstract, line 22. Clarify if this is a "lifetime" measure or "at last sex"
abstract line 22. please add % reporting they had ever used condoms and the % ever using oral contraceptives
abstract line 23. reword "resorted to the dual method" as "25.7% used both condoms and pills)
abstract line 24 and line 28 . not clear what comparison the p value refers to
line 24. Avoid the acronym LARC, and provide rates of use of IUDs separate from data on implant use
line 25. I do not understand how "recommended for use" is measured
line 31. sensitization is not the right word in English
The introduction (to line 78) should be condensed and edited by someone fluent in English
line 101-may not be necessary
line 102, section 2.5; was the analysis weighted for a complex population sampling strategy (as is done with the National Survey of Family Growth in the US?)
line 137, 169, 171, 182, 200, etc the p value is adequate with out the other greek letters; make sure to specify the comparison each p-value refers to
line 143, lifetime or last year mean number of sex partners?
line 149 "admitted" should be "reported". Sex without contraception is required when pregnancy is desired...
Table 2 and 3--lifetime, last year, or last sex?
line 178, "resorted" should be "consulted" [line 267 "resorted to" should be reported; line 280 again should not be resorted
Figure 5-is this among women who do not desire pregnancy or all women?
Table 4-the patch is not placed intravaginally; suggest clarifying "systemic hormonal (36%)" as you do not include the hormonal IUD in this group; I'm not familiar with the terms Ogino and billings [which are not used in the US...not sure if they are used elsewhere???]
line 272 I do not understand "implemented more intervals"
line 293, would add what year this study was collected
Author Response
Dear Reviewer 1, thank you for your valued observations and suggestions that were taken into account.
abstract, line 22. Clarify if this is a "lifetime" measure or "at last sex"
Response: Thank you, this suggestion was included.
abstract line 22. please add % reporting they had ever used condoms and the % ever using oral contraceptives
Response: Percentages were indicated following your indication
abstract line 23. reword "resorted to the dual method" as "25.7% used both condoms and pills)
Response: This sentence has been rewritten.
abstract line 24 and line 28 . not clear what comparison the p value refers to
Response: The p value refers to the last sentence. The previous sentences show data descriptively.
line 24. Avoid the acronym LARC, and provide rates of use of IUDs separate from data on implant use
Response: This data is included in the results section. Due to limitations of space, the first suggestion could not be amended.
line 25. I do not understand how "recommended for use" is measured
Response: When health providers carried out a counselling on contraception, only 50% included LARC as a choice to offer women.
line 31. sensitization is not the right word in English
Response: Thank you for the observation. This word was changed.
The introduction (to line 78) should be condensed and edited by someone fluent in English
Response: The text has been reviewed by an expert.
line 101-may not be necessary
Response: This has been eliminated.
line 102, section 2.5; was the analysis weighted for a complex population sampling strategy (as is done with the National Survey of Family Growth in the US?)
Response: Yes, it was.
line 137, 169, 171, 182, 200, etc the p value is adequate with out the other greek letters; make sure to specify the comparison each p-value refers to
Response: All p-value were reviewed, A note was included explaining the meaning of the Greek letters. Please, see lines 103-105.
line 143, lifetime or last year mean number of sex partners?
Response: This was modified, and it was specified that the information referred to the last year.
line 149 "admitted" should be "reported". Sex without contraception is required when pregnancy is desired...
Response: We have maintained the original expression because women did not always avoid contraception because they desired a pregnancy. Sometimes, they did not use them due to other motives.
Table 2 and 3--lifetime, last year, or last sex?
Response: This was clarified in table 2 and 3.
line 178, "resorted" should be "consulted" [line 267 "resorted to" should be reported; line 280 again should not be resorted
Response: All terms “resorted” were changed for “consulted”
Figure 5-is this among women who do not desire pregnancy or all women?
Response: This figure refers to all women.
Table 4-the patch is not placed intravaginally; suggest clarifying "systemic hormonal (36%)" as you do not include the hormonal IUD in this group; I'm not familiar with the terms Ogino and billings [which are not used in the US...not sure if they are used elsewhere???]
Response: This suggestion was included. All terms were reviewed.
line 272 I do not understand "implemented more intervals"
Response: This term was rephrased. It means that women stopped taking their contraception for a while to avoid hormones during that period.
line 293, would add what year this study was collected
Response: Thank you, this particular data was added.
Reviewer 2 Report
Dear Authors
The presented study tackles an issue of Results of the National Contraception Survey conducted by Sociedad Española de Contracepción (2020).The study was conducted reliably with appropriate selection of tests. The strenght of the study is a large sample size. Overall, some minor issues require complementary information:
1. I suggest including the information about legaly approved minimum age of sexual intercourse in Spain.
2. I suggest redrafting the Table 3- it’s a little unclear.
3. I suggest expanding the abbreviations used in the text (like LARCs and SARCs)
Author Response
Dear Reviewer 2, thank you for your valued observations and suggestions that were taken into account.
Dear Authors
The presented study tackles an issue of Results of the National Contraception Survey conducted by Sociedad Española de Contracepción (2020).The study was conducted reliably with appropriate selection of tests. The strenght of the study is a large sample size. Overall, some minor issues require complementary information:
- I suggest including the information about legaly approved minimum age of sexual intercourse in Spain.
Response: This has been included. Please, see line 129.
- I suggest expanding the abbreviations used in the text (like LARCs and SARCs)
Response: This was amended. Please, see line 54.